# Mouse Lockbox Dataset: Behavior Recognition of Mice Solving Mechanical Puzzles

## Abstract

Machine learning and computer vision have a major impact on the study of natural animal behavior, as they enable automated action classification of large bodies of videos. Mice are the standard mammalian model system in many fields of research, but the open datasets that are currently available to refine machine learning methods mostly focus on either simple or social behaviors. In this work, we present a large video dataset of individual mice solving complex mechanical puzzles, so-called lockboxes. The dataset consists of a total of well over 110 hours of animal behavior, recorded with three cameras from different perspectives. As a benchmark for frame-level action classification methods, we provide human-annotated labels for all videos of two different mice, that equal 13% of our dataset. The used keypoint (pose) tracking-based action classification framework illustrates the challenges of automated labeling of fine-grained behaviors, such as the manipulation of objects. We hope that our work will help accelerate the advancement of automated action and behavior classification in the computational neuroscience community. An anonymized preview of our dataset is available for the reviewers of this manuscript at `https://www.dropbox.com/scl/fo/h7nkai8574h23qfq9m1b2/AP4gNZOpDJJ7z0yGtbWQiOc?rlkey=w36jzxqjkghg0j0xva5zsxy2v&st=5r9msqjw&dl=0`

## 1 Introduction

Ethology, the study of non-human behavior, (Tinbergen, 1961) is one of the cornerstones of understanding complex biological systems. In recent years, with the integration of machine learning into the field, computational ethology (Anderson & Perona, 2014) emerged as a powerful new paradigm offering new pathways for advancing both fields and beyond. For instance, it has significantly influenced neuroscience, enabling the development of computational frameworks that bridge neural mechanisms with observations of behaviors (Datta et al., 2019; McCullough & Goodhill, 2021; von Ziegler et al., 2021; Kennedy, 2022). In robotics, animal behavior datasets allow researchers to learn artificial agents to navigate and interact autonomously in natural environments. The hypothesized learning models used in this process can then be tested by comparing the performance of the learned agents against that of natural agents (Baum et al., 2022). Furthermore, these datasets also provide a source of inspiration for developing machine learning approaches capable of handling high-dimensional, temporal, (Jia et al., 2022) and eventually multimodal data.

The available datasets of freely moving animals (Burgos-Artizzu et al., 2012; Dunn et al., 2021; Pedersen et al., 2020; Eyjolfsdottir et al., 2021; Marshall et al., 2021; Segalin et al., 2021; Sun et al., 2021a; Ng et al., 2022; Hu et al., 2023; Ma et al., 2023; Rogers et al., 2023; Zia et al., 2023; Brookes et al., 2024; Duporge et al., 2024; Kholiavchenko et al., 2024; Li et al., 2024) provide the foundation for the development of automated behavioral analysis tools, e.g., B-SOiD (Hsu & Yttri, 2021), VAME (Luxem et al., 2022), and Keypoint-MoSeq (Weinreb et al., 2024). However, all of these datasets and their descending methods focus on trivial and social behaviors, but neglect the structure imposed by well-defined tasks that provoke complex behaviors. This absence limits their applicability for studying goal-directed actions, problem-solving, and other behaviors critical to understanding cognitive processes in neuroscience, robotics, and artificial intelligence.

Action classification is central for understanding behavior. For instance, based on a sequence of actions researchers can analyze whether an animal has "understood" a task as it follows a policy

that advances it towards a goal. Scientists can also study learning by focusing on policy changes or by trying to infer goals, e.g., by the means of inverse reinforcement learning. Doing so requires unbiased modeling of sequential data, identifying (unknown) patterns, and making predictions in noisy, real-world environments. As of today, the state-of-the-art approaches in computational ethology (Hsu & Yttri, 2021; Luxem et al., 2022; Weinreb et al., 2024) build upon predefined keypoints. However, this may make meeting the stated requirements challenging as keypoints ignore possibly high descriptive visual information other than location. Therefore, the field is in need of robust representation learning that generates expressive features for complex behavioral data. They can help capture the high-dimensional structure of actions and behaviors, offering generalizable insights that are transferable across both tasks and species.

In this work, we provide the first large-scale labeled, single-agent, multi-perspective video dataset of mice showing intelligent behavior as they learn to solve mechanical puzzles, so-called lockboxes. Every lockbox consists of a single or a combination of four different mechanisms, which can only be solved by a specific sequence, and is baited with a food reward. Once a mouse succeeds to open a lockbox, it gains access to the food reward. To provide a benchmark for novel representation learning methods, we provide labels for 13% of the video data, including mechanism state, mouse-to-mechanism proximity, and both mouse-mechanism and mouse-reward actions. This amounts to about 15 hours and 25 minutes, i.e., more than 1.6 million frames. In doing so, we increase the longest total video playtime, i.e., the number of perspectives multiplied by the real time recorded, available through any dataset showing mice from 88 hours (Burgos-Artizzu et al., 2012) before by more than 33% to now 117 hours and 52 minutes.

To guarantee a high quality of labeled data, each labeled video is annotated by two skilled human raters who have been instructed prior to annotating. The consistency between raters is assessed by their inter-rater reliability (McHugh, 2012), providing an objective and well-established measure of agreement. We regard such rigorous and transparent annotation protocols as essential for creating datasets that allows assessing the performance of future machine learning approaches.

We use a keypoint-based approach as an initial benchmark for our dataset (Boon et al., 2024) that aligns with the well-established three-parted pipeline introduced by (Anderson & Perona, 2014), i.e., animal tracking, action classification, and behavioral analysis. Furthermore, we compare our human-human agreement against its human-machine agreement. In the absence of established benchmark methods for the interaction of natural agents with their environment, this will allow others to assess the performance of their approaches.

In summary, we contribute a new, multi-perspective, video dataset that consists of mice learning to solve lockboxes. We hope that our dataset will serve three purposes. First, we hope that it will promote the advancement and adoption of more diverse machine learning approaches in computational neuro-/ethology. Second, it may provide interesting challenges to the representation learning community, as behavioral action classification requires both large-scale pose and fine-level visual information, e.g., the position of mouth and teeth. And third, we hope that a broader analysis of the dataset by the research community will advance our understanding of how natural agents learn to solve complex problems.

## 2 RELATED WORK

The general three-parted structure of automated behavioral analysis—animal tracking, i.e., localization of keypoints (poses) of individual animals and tracking them over time; action classification, i.e., identification of time intervals when relevant action patterns are performed; and behavior analysis, i.e., estimating behavioral patterns assembled from sequences of actions—(Anderson & Perona, 2014) largely persists in state-of-the-art approaches (Datta et al., 2019; von Ziegler et al., 2021; Kuo et al., 2022; Luxem et al., 2023; Fazzari et al., 2024), albeit with increasingly advanced methods. It is the most common approach to first detect animal poses (Mathis et al., 2018; Alameer et al., 2020; Dunn et al., 2021; Brattoli et al., 2021; Segalin et al., 2021; Pereira et al., 2022; Russello et al., 2022; Biderman et al., 2024) and further process them to trajectories (Alameer et al., 2020; Hsu & Yttri, 2021; Segalin et al., 2021; Sun et al., 2021b; Luxem et al., 2022; Biderman et al., 2024; Boon et al., 2024) or feature representations (Brattoli et al., 2021; Zhou et al., 2023), while only few of the available works (Batty et al., 2019; Bohnslav et al., 2021; Brattoli et al., 2021; Jia et al., 2022) shift towards encoding videos as abstract spatiotemporal features. Both pose trajectories (Alameer et al.,

Table 1: Overview of some distinguishing properties of available video datasets showing rodents. The listed durations, i.e., real time recorded and calculated total playtime, are rounded values. The 20 (intelligent) behaviors we report reflect the composition of five labeled interactions that the mice may perform on the four distinct lockbox mechanisms.

| | CONTEXT | LABELS | # PERSPECTIVES $\times$ REAL TIME |
|---|---|---|---|
| **CRIM13** | Mice, social | 13 (social) behaviors | $2 \times 44\text{h} \approx 88\text{h}$ |
| **Rat 7M** | Rats, individual | 20 keypoint markers | $6 \times 11\text{h} \approx 65\text{h}$ |
| **PAIR-R24M** | Rats, social | 14 (social) behaviors | $24 \times 9\text{h} \approx 220\text{h}$ |
| **MARS** | Mice, social | 3 social behaviors | $2 \times 14\text{h} \approx 28\text{h}$ |
| **CalMS21** | Mice, social | 3 social behaviors | $1 \times 70\text{h} \approx 70\text{h}$ |
| **Ours** | Mice, individual | 20 (intelligent) behaviors | $3 \times 40\text{h} \approx 120\text{h}$ |

2020; Brattoli et al., 2021; Hsu & Yttri, 2021; Segalin et al., 2021; Sun et al., 2021b; Luxem et al., 2022; Biderman et al., 2024; Weinreb et al., 2024) as well as abstract spatiotemporal features (Batty et al., 2019; Bohnslav et al., 2021; Brattoli et al., 2021; Jia et al., 2022) then form the basis for the next analysis steps, the quantification of actions and behaviors.

To refine these methods, various video datasets are available to the community today. We limit the following overview to those that show rodents, because various rodent species can potentially be used in domain transfer settings, due to their largely similar visual appearance and motor apparatus. Table 1 summarizes some of their distinguishing properties discussed below. A full survey on (both video and image) datasets showing animals would substantially exceed the scope of this work.

Burgos-Artizzu et al. (2012) presented with CRIM13 the up to now largest dataset with a total of 88 hours (44 hours of recorded real time) of video data showing mice from two (top-down and side) perspectives in resident-intruder contexts. They provide 13 human-annotated (social) behavior labels—approach, attack, coitus, chase, circle, drink, eat, clean, human, sniff, up, walk, and other—for each of the 237 pairs of 10 minute long videos. For these labels, they report a 70% agreement among human raters while the method they propose reaches 61.2% human-machine agreement for behavior classification.

Dunn et al. (2021) presented Rat 7M, a dataset consisting of 65 hours (11 hours of recorded real time) worth of videos of rats with 20 markers pierced to their bodies. The rats were recorded using six cameras, and 12 motion capture cameras were used to record the markers' coordinates in space. Behavior labels are not provided. They report that the pose tracking approach they proposed is robust in domain transfer settings where the species of the tracked agent changes from rat to mouse.

Marshall et al. (2021) presented PAIR-R24M, a dataset consisting of 220 hours (9 hours of recorded real time) worth of videos of rats from 24 perspectives. They provide 14 human-annotated (social) behavior labels—amble, crouch, explore, head tilt, idle, investigate, locomotion, rear down, rear up, small movement, sniff, groom, as well as close to, explore, and chase—for the entire dataset. It is the most perspective-diverse, the largest by total playtime, but also the shortest by real time recorded.

Segalin et al. (2021) presented MARS, a dataset consisting of 28 hours (14 hours of recorded real time) worth of videos of mice from two (top-down and front) perspectives. They provide three human-annotated social behavior labels—attack, investigation, and mount—for 3 hours (1.5 hours in real time) worth of video data in 10 videos. They do not only propose a method that reaches human-level performance in behavior classification but also a graphical user interface that will accelerate computer-aided research in neuroscience labs that do not employ machine learning experts.

Sun et al. (2021a) presented CalMS21, a 70 hour long video dataset showing pairs of mice from a top-down perspective. They provide three human-annotated social behavior labels—attack, investigate, and mount—for 10 hours worth of video data.

## 2.1 Benchmark Method

Since methods based on keypoint (pose) estimation and tracking are currently state-of-the-art, our benchmark experiments are based on the pose-tracking approach used by Boon et al. (2024). The method consists of 3 steps: the use of DeepLabCut (DLC) for 2-dimensional pose tracking, 3-dimensional reconstruction and the refinement of keypoint data using (Extended) Kalman filtering, and the detection of action labels. A high-level description of steps is given below.

First, 2-dimensional poses of the mice and lockbox mechanisms are extracted from the videos on a frame-level by learning DLC models under supervision. We learn one DLC model to locate keypoints of mice, and two that locate keypoints of lockbox mechanisms—one for the single-mechanism lockboxes, and one for the lockbox combining them—using default parameters (Mathis et al., 2018; Nath et al., 2019) (see Appendix A.2). Next, the scene is reconstructed by utilizing the known 3-dimensional locations of the lockbox mechanisms given by their CAD models. We linearly map the known 3-dimensional locations onto the corresponding triplets of 2-dimensional keypoints using the random sample consensus (RANSAC) algorithm and construct a triangulation matrix for each video. Each of these triangulation matrices is then used as an observation matrix for a(n) (Extended) Kalman filter to refine the observed triplet of 2-dimensional keypoints into a common 3-dimensional space. The head and the tail of the mouse are inferred using a skeletal model, while the keypoints of the mechanisms and the paws of the mouse are inferred as single keypoints. Finally, the interactions of the mice with the lockbox mechanisms are detected based on the 3-dimensional poses of the mouse and predefined bounding boxes spanned by the 3-dimensional keypoint locations. For the proximity labels, the snout of mouse is used to detect the actions: each frame in which the snout of the mouse is inside of a bounding box, the corresponding action label (e.g., proximity lever) is detected. Biting is detected using the mouth of the mouse, which location is computed from the rigid body model of the mouse head. And the touch labels are deteced using the locations of the front paws. Note that the bite and touch labels have different predefined bounding boxes than the proximity labels, as these actions have a finer level of granularity than proximity labels.

# 3 Dataset

In this section, we describe our dataset in detail. This includes a description of the mice, the arena as part of the home cage and schematics of the lockboxes that the mice are presented with, the camera setup, the schedule at which mice were presented with the lockboxes, the preprocessing of the recorded videos in order to refine them to a dataset suitable for computer vision and machine learning approaches, the annotation of behavior labels including our ethogram, statistics on videos and labels, benchmark results, and known limitations.

## 3.1 Data Collection and Preprocessing

To create this video dataset, 12 female C57BL/6J mice obtained from Charles River Laboratories (Sulzfeld, Germany) were recorded in a free-standing Makrolon type III cage, that was connected to another cage of the same type by a tube. The mice were housed in groups of 4 animals in a 12/12-hour light/dark cycle of artifical light. During the trials with the lockboxes that took place in light phases, only one animal at a time could enter the cage in which the lockboxes were presented. The cage was closed with a top grid that was partially removed (cutout) to allow for unobstructed view on the lockbox. Three Basler acA1920-40um cameras (LM25HC7 lens, f = 25mm, k = 1.4; Kowa, Nagoya, Japan) were setup to record the grayscale videos at a 1936×1216px resolution (the highest possible) with a 30fps frame rate. Additionally, we used two infrared lights (Synergy 21 IR-Strahler 60W, ALLNET GmbH Computersysteme, Germering, Germany) to illuminate the cage. The advantages of infrared lights were that they enhanced the quality of recordings captured by the infrared-sensitive cameras we used, while also not being aversive to the animals.

Figure 1a depicts the setup described before. All cameras were connected to a single computer and controlled by a common software program to synchronize frame capturing. The mice were presented with five different lockboxes: a combined lockbox consisting of four interlocked mechanisms (Figure 1b), and four simpler lockboxes presenting these mechanisms individually (Figure 1c). A hidden food reward (oatmeal flake) was used to bait the mice to solve the lockbox. It is important to note that the mice were not subjected to food or water deprivation. They had ad libitum access to food

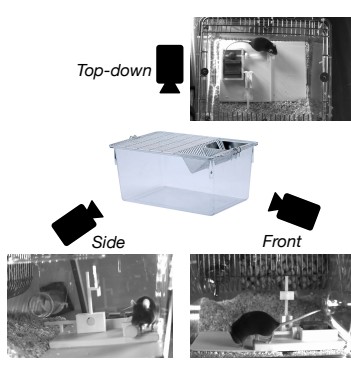
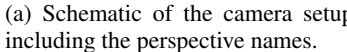

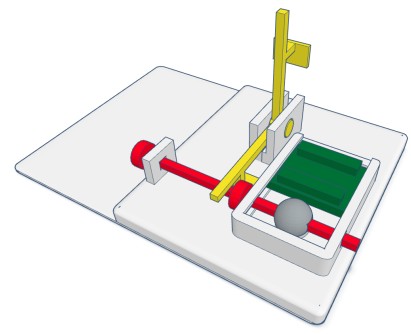

(a) Schematic of the camera setup including the perspective names.

(b) Lockbox of combined mechanisms baited with a food reward underneath the sliding door.

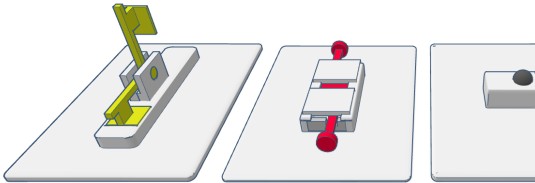

(c) Single-mechanism lockboxes baited with a food reward underneath each mechanisms.

Figure 1: Camera setup used for recording the videos, as well as lockboxes and their mechanisms: lever (yellow), stick (red), ball (gray), and sliding door (green). Each lockbox is baited with a food reward underneath the (last) mechanism. Appendix A.1 provides figures of the lockboxes with unlocked mechanisms.

pellets (LASvendi, LAS QCDiet, Rod 16, autoclavable) and tap water. Therefore it can be assumed that they were not hungry when entering the arena. However, the food reward was exclusively provided within the lockboxes. To familiarize the mice with the food reward, they were habituated over three consecutive days prior to the start of lockbox training by placing eight oat flakes at the location where the lockbox would be introduced during the training sessions. The freely behaving mice were presented with the combined lockbox for at total of 6 and with the single-mechanism lockboxes 11 trials. In each trial, the mice were first exposed to the combined lockbox followed by a randomized order of single-mechanism lockboxes. The videos end shortly after the reward is reached, or if a trial reached the maximum duration of 30 minutes for combined and 15 minutes for single-mechanism lockboxes.

We manually cut the videos to remove disturbances, such as the experimenter's hands switching lockboxes. Any videos where the lockboxes could not be seen entirely were filtered out. This resulted in a dataset with a total playtime of 117 hours and 52 minutes.

## 3.2 LABEL ANNOTATION

We provide human annotations of the mechanism state, mouse-to-mechanism proximity, and both mouse-mechanism and mouse-reward action labels. To do so while also preventing any kind of information leakage between labeled and unlabeled data splits, we labeled all videos of two specific mice (mouse numbers 291 and 324) that have a combined total playtime of about 15 hours and 25 minutes in 270 videos, i.e., more than 1.6 million frames in 90 trials. This equals about 13% of our dataset's total size.

Table 2 defines the ethogram we used to instruct our nine skilled human raters. Appendix A.3 provides example frames for the different labels. We used these labels that express trivial truths in order to minimize anthropomorphic biases, that would otherwise distort the evaluation of experiments and the conclusions drawn from their results. These biases are especially apparent when

Table 2: Ethogram used for label annotation.

| LABEL | DEFINITION |
|---|---|
| Proximity | The mouse's snout is within a distance of 1cm to a specific mechanism. |
| Touch | The mouse touches a specific mechanism with one or both of its front paws. |
| Bite | The mouse bites into a specific mechanism. |
| Unlock | The state of a specific mechanism changes to unlocked. This may make the reward accessible or enabling the next mechanism to be unlocked. State changes may occur without the mouse manipulating a mechanism directly. |
| Lock | The state of a specific mechanism changes to locked. This may make the reward inaccessible or preventing the next mechanism from being unlocked. State changes may occur without the direct manipulation of a mechanism. |
| Reach reward | The mouse is in first contact with the reward with any of its body parts. |

using more high-level labels, such as exploring and deliberately manipulating lockbox mechanisms, that strongly depend on subjective human interpretation. Using more explicit labels not only leads to higher label quality but also lowers the risk of computer vision and machine learning models learning said biases before reintroducing them as noise to any analysis based on their outputs.

For annotating the labels, we merged every video triplet (top-down, side, and front perspective) into a combined video.[1] All labels have been annotated by a random pair of raters with a temporal accuracy of $\pm 100$ milliseconds, i.e., $\pm 3$ frames using BORIS (Friard & Gamba, 2016). It took each of our raters about 6.2 to 11.5 times longer than the actual playtime to annotate the labels in a video. This matches with the factor of 5 to upmost 10 that is reported throughout the available literature. We account our slightly higher efforts to the multitude of mouse body parts and lockbox mechanisms that needed to be observed at the same time.

### 3.3 DATASET STATISTICS

In this section, we give an overview over various data statistics for both the labeled and unlabeled videos. It is worth mentioning that the unevenly distributed playtime shares of different mechanisms as well as active labels is rooted in the mice behaving freely in the arena. Their inherent preference for different actions and mechanisms is naturally occurring and reflected in the statistics we report.

### 3.3.1 PLAYTIME STATISTICS

Our dataset has a total playtime of 117 hours and 52 minutes, i.e., almost 13 million frames, that show 39 hours and 17 minutes of real experimental time recorded from 3 perspectives. The dataset consists of a total of 1629 videos, i.e., 543 trials. Table 3 gives a detailed overview of the playtime shares for both mice and lockbox mechanisms. Figure 2 shows a histogram of videos playtimes. The videos in our dataset have a mean playtime of 4 minutes and 21 seconds.

### 3.3.2 LABEL STATISTICS

We provide human-annotated mechanism state, mouse-to-mechanism proximity, and both mouse-mechanism and mouse-reward action labels for mouse numbers 291 and 324, to avoid information leakage between labeled and unlabeled data splits. This totals to 15 hours and 25 minutes, i.e., more than 1.6 million frames of video data, as Table 3 shows.

Figure 3 shows the inter-rater reliability, i.e., Cohen's kappa coefficients, (McHugh, 2012) for all pairs of human raters. On average our human raters annotate almost all proximity and touch labels

---

[1]Merging the video triplets into combined videos was necessary as BORIS version 8.27 suffers from a software issue that occurs more frequently when using it with multiple videos opened at once, and that causes to the software to crash only minutes into using it. The published dataset does not include the merged videos.

Table 3: Playtime shares of both different mice and mechanisms in our dataset in percent. The column names identify the mice while the rows specify the mechanisms.

| | 52 | 68 | 70 | 80 | 162 | 192 | 258 | 285 | 291 | 324 | 336 | 389 | $\sum$ |
|---|---|---|---|---|---|---|---|---|---|---|---|---|---|
| **Lever** | 1.0 | 1.9 | 0.8 | 1.8 | 2.0 | 0.7 | 2.3 | 0.3 | 1.4 | 1.0 | 0.4 | 0.6 | 14.2 |
| **Stick** | 0.9 | 1.1 | 1.2 | 1.0 | 1.1 | 0.5 | 0.7 | 0.4 | 0.9 | 0.5 | 0.5 | 1.4 | 10.1 |
| **Ball** | 0.6 | 0.8 | 0.6 | 0.9 | 2.3 | 1.4 | 0.5 | 0.4 | 0.8 | 0.3 | 0.8 | 0.4 | 9.7 |
| **Sl.Door** | 1.3 | 3.6 | 2.0 | 0.7 | 0.5 | 0.5 | 1.1 | 0.4 | 0.4 | 0.3 | 2.7 | 0.5 | 13.9 |
| **Comb.** | 3.2 | 7.6 | 4.9 | 3.6 | 3.1 | 4.6 | 3.9 | 3.8 | 2.4 | 5.2 | 6.0 | 3.9 | 52.0 |
| $\sum$ | 6.9 | 15.1 | 9.4 | 7.9 | 9.0 | 7.7 | 8.5 | 5.3 | 5.8 | 7.3 | 10.3 | 6.7 | **100** |

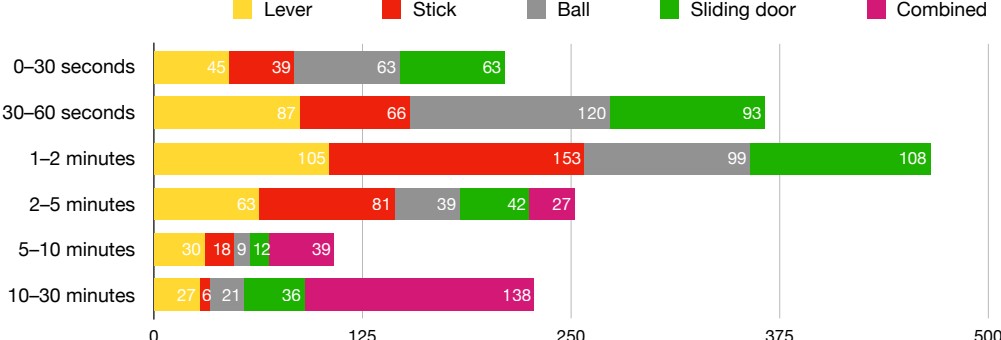

Figure 2: Histogram of the video playtime distribution with pseudo-logarithmically scaled bins. The lower limits of the bins are excluded while the upper limits are included, and the different mechanisms are color coded. The different lockbox mechanisms are color coded.

with a moderate or even strong agreement, but have a lower agreement for the stick mechanism. In contrast, they annotate bite labels with only minimal to weak agreement. We account this to the bite label being particularly hard to annotate as it is not always directly visible in the videos.

Table 4 shows the playtime shares of different action label classes. It gives an overview of the density of active behavior labels for the different lockbox mechanisms relative to the total labeled playtime. It furthers gives the density of either behavior label being active for any of the mechanisms.

Table 4: Playtime shares of different action labels relative to the total playtime of the labeled videos in percent.

| | Lever | Stick | Ball | Sl.Door | Any |
|---|---|---|---|---|---|
| **Proximity** | 15.73 | 19.05 | 13.41 | 18.97 | 55.39 |
| **Touch** | 7.06 | 4.07 | 7.00 | 9.32 | 25.50 |
| **Bite** | 1.81 | 1.50 | 3.41 | 1.42 | 8.12 |

## 3.4 BENCHMARK RESULTS

Next to manually annotating the trials of two mice, we used our keypoint tracking pipeline to automatically generate labels on a frame-to-frame basis, which are used here as a benchmark method. The trials of the two mice are considered to be the test set for our benchmark method and are therefore not used in its training procedure. Analogous to the inter-rater reliability of the previous section, we compare the resulting action labels from our benchmark to both human raters in Figure 3.

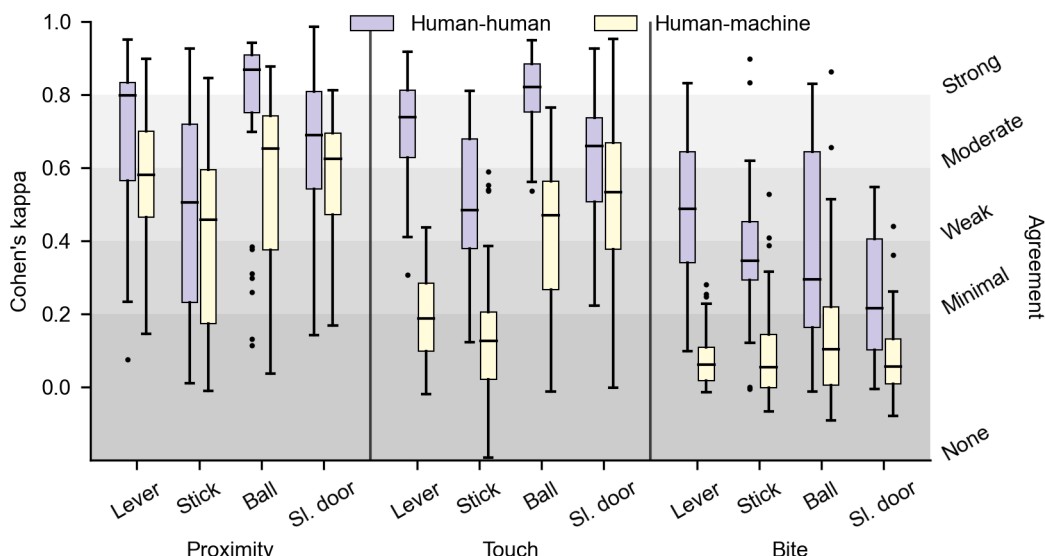

Figure 3: Inter-rater reliability measured using Cohen's kappa coefficients, to assess both human-human and human-machine agreement in label annotation for both different action classes and mechanisms. The human-human inter-rater reliability is colored purple while the human-machine inter-rater reliability is colored yellow.

The benchmark method performs well for proximity labels. This becomes apparent when comparing the human-machine against the human-human inter-rater reliability, where our benchmark method mostly reaches human-level performance. In contrast, for both touch and bite labels it is outperformed by our human raters. These two action labels require a higher accuracy in the detection of the pose of the mouse as well as the reconstruction of the bounding box of the mechanisms. Therefore, the reliability for touch and bite labels are naturally lower than for proximity.

Interestingly, the proximity and touch action labels for both the ball and sliding door have a higher inter-rater reliability than the lever and the stick. We assume that this difference originates from the ball and the sliding door mechanisms being more easily approximated by bounding boxes than the lever and the stick.

## 3.5 LIMITATIONS

Our dataset has three limitations. First, since the video recording was pseudo-synchronized by our recording software, the frames of different cameras have been captured with a temporal desynchronization. We sampled the average asynchronicity to be 1.39 frames with a standard deviation of 1.50 frames. We do not expect this to cause any issues other than in settings that would, e.g., require 3-dimensional keypoints to be tracked with an accuracy much higher than the accuracy we annotated our labels with. Second, not all videos share the same exact positioning of the cameras as the videos have been recorded over the course of several months so our setup had to be rearranged over time. And third, due to technical issues during the data acquisition, i.e., insufficient lighting conditions and severe camera dislocation, some trials had to be discarded from the dataset which lead to an imbalanced number of videos per mouse.

## 4 CONCLUSION

In this work, we presented the—to the best of our knowledge—first available single-agent, multi-perspective video dataset of mice showing intelligent behavior as they learn to solve mechanical puzzle mechanisms. These so-called lockboxes consist of either one of four mechanisms or their combination, and are baited with a food reward. As a benchmark for novel approaches, we provide a range of human-annotated labels—the mechanism states, the proximity of a mouse to a mechanism,

if a mouse is touching or biting a mechanism, and when the mouse reaches the food reward—for 13% of our 117 hours and 52 minutes long video dataset. This equals an increase of over 33% in total video playtime available through any mouse dataset available today.

As an initial comparison of human annotations with automated methods, we provide labels generated from a state-of-the-art keypoint-based pose tracking approach as a benchmark method. We compare the human-human against the human-machine inter-rater reliability and find that the automatic detection of the proximity of a mouse to the lockbox mechanisms can be considered robust, while the more fine-grained action labels touching and biting require more precise keypoint localization rendering the benchmark results unreliable. However, since these labels are indispensable for studying the complex behavior of an animal and to understand how this contributes to learning, we are convinced that approaches beyond keypoint (pose) tracking, e.g., representations learnt without any or under self-supervision, are crucial to future advancements in neuroscience. We hope that our dataset will contribute to this advancement by challenging and inspiring others.

An anonymized preview of our dataset is available for the reviewers of this manuscript at `https://www.dropbox.com/scl/fo/h7nkai8574h23qfq9m1b2/AP4gNZOpDJJ7z0yGtbWQiOc?rlkey=w36jzxqjkghg0j0xva5zsxy2v&st=5r9msqjw&dl=0`

## ACKNOWLEDGMENTS

We thank our encouraged lab assistants for their support with cleaning the raw video data and annotating the labels. Their dedication and hard work were essential to composing the presented dataset.

This project was funded by the Deutsche Forschungsgemeinschaft (DFG, German Research Foundation).

## ETHICS STATEMENT

Our research did not involve human subjects, sensitive data, harmful insights, nor methodologies or applications that may raise ethical concerns.

For generating the dataset underlying the present article, videos were recorded from 12 female C57BL/6J mice. Animals were at the age of 9 to 12 weeks when the videos used for the present article were recorded. Animal research was conducted in compliance with the local laws and regulations on the protection of animals used for scientific purposes.

The authors declare that they have no conflicts of interest. No sponsorships influenced this research.

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

# A  APPENDIX

## A.1  LOCKBOXES WITH UNLOCKED MECHANISMS

Figure 4 shows the opened lockboxes with symbolized food baits; see Figures 1b and 1c for reference.

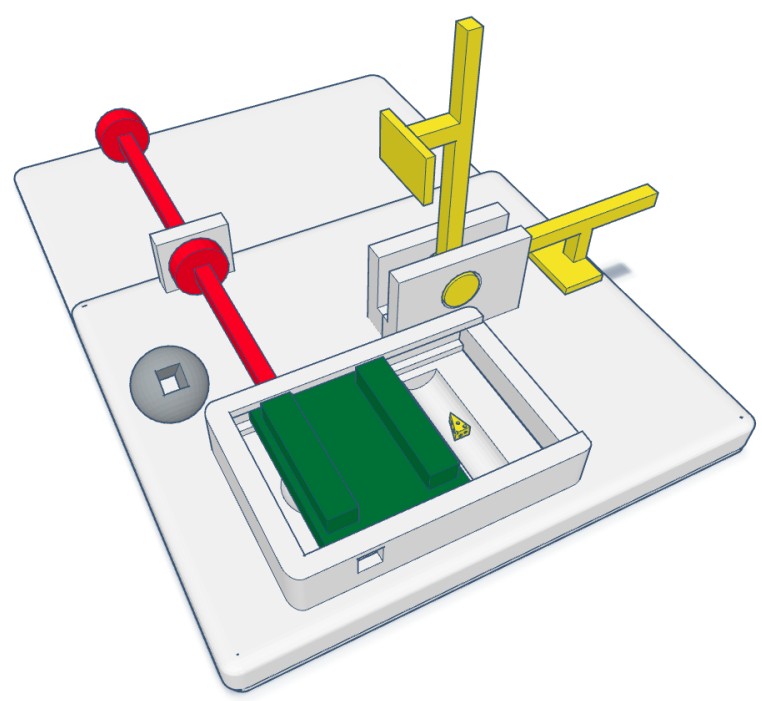

(a) Unlocked lockbox of combined mechanisms baited with a symbolized food reward underneath the sliding door.

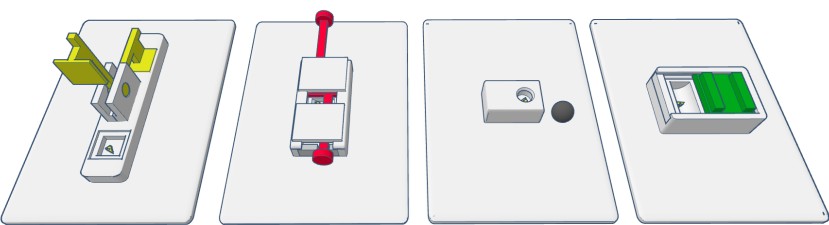

(b) Unlocked single-mechanism lockboxes baited with a symbolized food reward underneath each mechanisms.

Figure 4: Unlocked lockboxes and their mechanisms: lever (yellow), stick (red), ball (gray), and sliding door (green). This depiction contains symbolized food baits.

## A.2  KEYPOINT TRACKING

The DLC trackers are trained using human-annotated frames from the videos for which no action labels are available. The test sets of the trackers consist of labeled frames from the videos for which action labels are available (i.e. mouse 324 and 291).

Figure 5 shows examples of the keypoints used for training a DLC model that tracks the 2-dimensional locations of both a mouse and the lockbox mechanisms.

Figure 5: Examples of the keypoints used for tracking mice and lockbox mechanisms.

Table 5: The training and test errors for the keypoints used for mouse-tracker and the lockbox-trackers using DLC. The number in brackets represent the test errors for which the confidence of the tracker was above a threshold value of 0.6

| Mouse tracker | Training | Test |
|---|---|---|
| nose | 9.4 | 45.2 (7.8) |
| ear_left | 14.0 | 20.1 (18.5) |
| ear_right | 11.8 | 25.1 (18.1) |
| tail_base | 4.8 | 17.1 (8.1) |
| front_paw_left | 74.2 | 87.1 (8.7) |
| front_paw_right | 54.9 | 102.6 (30.2) |
| back_paw_left | 65.6 | 66.1 (70.0) |
| back_paw_right | 50.6 | 75.0 (69.6) |

| Combined lb | Training | Test | Single lb | Training | Test |
|---|---|---|---|---|---|
| lever_tip | 3.6 | 20.9 (5.6) | lever_tip | 8.1 | 5.3 (5.3) |
| other_lever_tip | 3.6 | 68.8 (39.9) | other_lever_tip | 3.2 | 100.8 (93.1) |
| stick_head | 3.6 | 7.7 (7.2) | stick_head | 2.1 | 52.3 (52.3) |
| ball | 3.6 | 18.7 (7.3) | ball | 2.4 | 5.6 (5.6) |
| sliding_door | 3.5 | 25.3 (11.7) | sliding_door | 2.4 | 110.7 (6.0) |

The training and test error (RMSE of the xy-coordinates in pixels) of the DLC trackers are shown in Table 5. In addition to outputting the locations of the keypoints, DLC additionally provides a confidence score between 0 and 1 for its predictions. This is often used for further analysis, for example by filtering certain predictions before using the keypoints as input to a Kalman filter. To provide a better idea on how the confidence influences the error, we additionally provide the RMSE for the test set at a threshold value of 0.6 (in brackets).

We have published the DLC tracks we created alongside our dataset.

### A.3 EXAMPLE FRAMES FOR LABELS

Figure 6 shows a selection of examples for our different label classes.

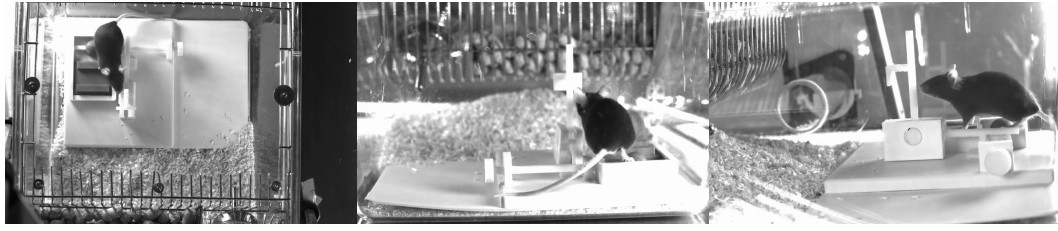

(a) Frame example with mouse in proximity to lever and touching the sliding door.

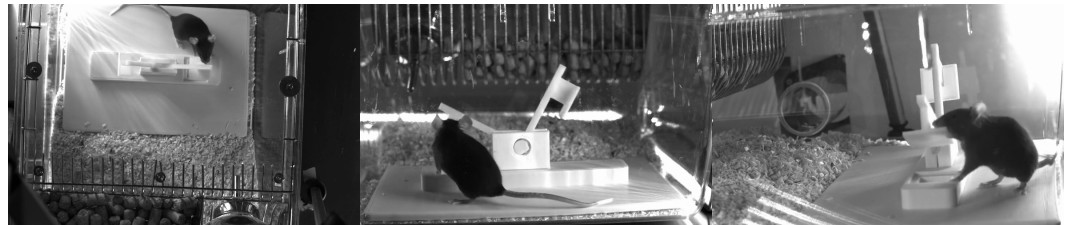

(b) Frame example with mouse in proximity to and biting the lever.

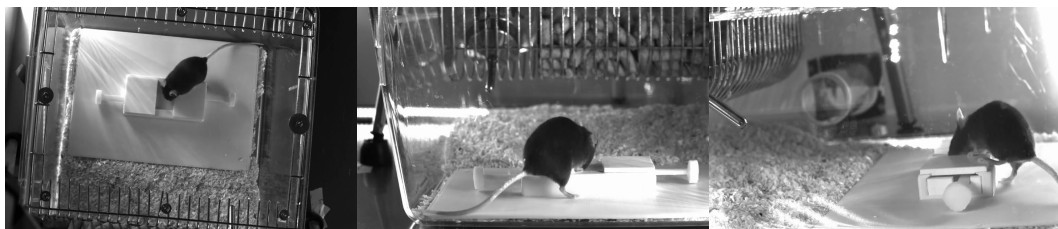

(c) Frame example with mouse in proximity to the stick.

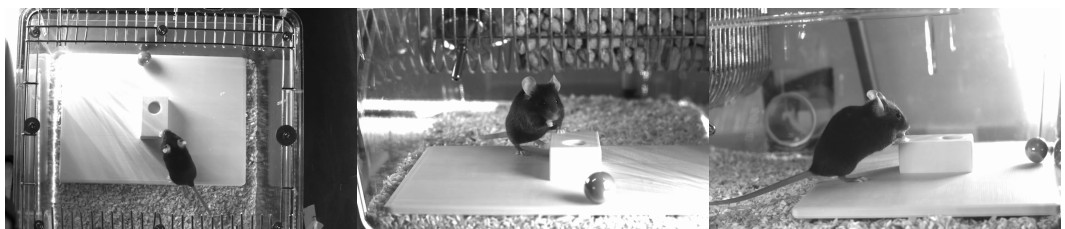

(d) Frame example with no action label active while the ball mechanism is unlocked.

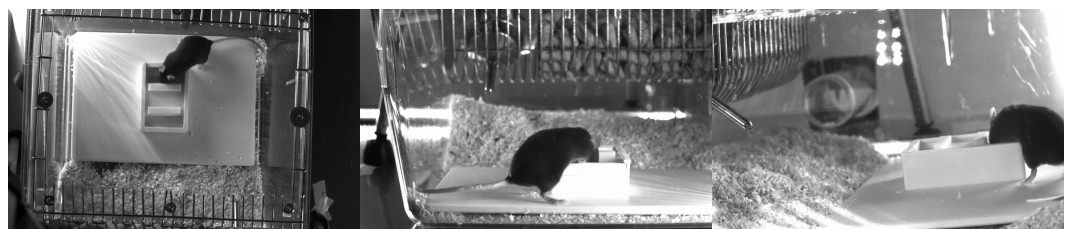

(e) Frame example with mouse in proximity to the sliding door while the sliding door mechanism is unlocked.

Figure 6: Example frames from labeled videos showing mice performing different actions.

## A.4 Disclosure of Our Approach to Literature Research

We have decided to silently add this section to our appendix as we consider it good practice to disclose all aspects of a scientific work, and we hope that it is useful to aspiring scientists.

For our rigorous literature research we mainly relied on the Google Scholar (`https://scholar.google.com`) and Semantic Scholar (`https://www.semanticscholar.org`) search engines using keywords and phrases relevant to our work. To further bolster the reliability of our literature research, we adopted a Markov blanket-like search pattern: for all of our references that we consider central to our work, we have filtered for further relevant work among their references, citations, and—depending on the context—both the references and citations of their citations. This allows us to search a highly contextualized corpus of several thousand publications in a structured, semantically meaningful, and thereby laborsaving way, significantly decreasing the risk of missing any relevant work.

