# OpenReview forum: "Mouse Lockbox Dataset: Behavior Recognition of Mice Solving Mechanical Puzzles"
_ICLR.cc/2025/Conference — Submitted to ICLR 2025_

### Official Review · Reviewer_FE9J · 2024-10-28

**Soundness:** 2
**Presentation:** 2
**Contribution:** 2
**Rating:** 6
**Confidence:** 3

**Summary:**

This paper collected a multi-perspective video dataset of individual mice solving complex mechanical puzzles. This paper also provided the human-annotated labels, i.e., the proximity between a mouse and a mechanism. The authors give detailed statistics of playtime shares per lockbox.

**Strengths:**

1. This paper collected a large-scale video dataset of individual mice solving complex mechanical puzzles.
2. This paper is well-organized and easy to read.
3. This paper provides a comprehensive review of relevant work.

**Weaknesses:**

1. This paper only collected a new dataset and ignored benchmark methods for recognizing mouse actions.
2. The comparison between the dataset collected in this paper and previous datasets is unclear. The author can provide a summary table to illustrate the differences from previous work.
3. There is no baseline or proposed method regarding the behavior recognition of mice.

**Questions:**

1. What pose keypoint definitions were used in this work? Will different key point definitions affect recognition?
2. Are all the mice in the dataset presented in this paper under normal light conditions?
3. Does the hunger state of mice affect their ability to solve mechanical puzzles?

---

> ### Author Response · Authors · 2024-11-21
>
> Dear reviewer, we are thankful for your insightful feedback as it allowed us to improve our manuscript. Please allow us to address both the weaknesses and questions you stated in your review.
>
> Any changes that we have introduced to the text of our revised manuscript have been marked with blue color.
>
> As weaknesses 1 and 3, you argue that we ignored existing benchmark methods and did not provide a baseline. While there exist methods for behavior identification on individual mice (e.g., Keypoint-MoSeq [1], B-SOiD [2], and VAME [3]), these methods extract behavioral primitives of a mouse in isolation, rather than interactions with its environment as we do here. It is nontrivial to extend them such that they couple an action (e.g., touch) with an environmental element (e.g., a lockbox mechanism) and there is currently no established standard that could serve as a benchmark. Therefore, we decided against using these existing methods as a benchmark for our dataset. Instead, we applied an established keypoint tracking method (i.e., DeepLabCut [4]) to extract the 2D keypoints from the videos and implemented a generic processing pipeline, based on (Extended) Kalman filters, to detect proximity, touching, and biting in 3D space. We describe this method and evaluate its results in sections 2.1 and 3.4.
>
> As weakness 2, you argue that the comparison between our and other datasets is unclear and suggest to give an overview in a table. We have added such a table (Tbl. 1).
>
> In question 1, you ask for the definition of the keypoints used for tracking mice and mechanisms. We have added a figure (Fig. 5) and a table (Tbl. 5) to our appendix A.2 showcasing the keypoints we used.
>
> In question 2, you asked if the videos were recorded under normal light conditions. The mice were kept in a 12/12-hour light/dark cycle with artificial (infrared) lighting. We have added this and some additional information in the first paragraph of section 3.1.
>
> In question 3, you asked if the hunger state of the mice affected their ability to solve the lockboxes. We cannot answer this question because the hunger state was not varied systematically. The mice had ad libitum access to both food pellets and tap water in the home cages. Therefore it can be assumed that they were not hungry when entering the arena. The food reward was exclusively provided in the arena. Before conducting the experiments, the mice have been familiarized with the food reward. We have added this information to section 3.1, lines 251–257.
>
> We hope that we were able to properly address your questions and remarks, and appreciate additional feedback.
>
> [1] Weinreb et al. Keypoint-MoSeq: parsing behavior by linking point tracking to pose dynamics. Nature Methods, vol. 21, no. 7, pp. 1329–1339, 2024.
>
> [2] Hsu et al. B-SOiD: an open source unsupervised algorithm for discovery of spontaneous behaviors. BioRxiv, 770271, 2019.
>
> [3] Luxem et al. Identifying behavioral structure from deep variational embeddings of animal motion. Communications Biology, vol. 5, no. 1, pp. 1267, 2022.
>
> [4] Mathis et al. DeepLabCut: markerless pose estimation of user-defined body parts with deep learning. Nature Neuroscience, vol. 21, pp. 1281–1289, 2018.

---

> > ### Comment · Reviewer_FE9J · 2024-11-23
> >
> > Dear authors, thank you for your response. My concerns have been addressed.
> > After reading the comments from other reviewers, I am willing to raise my rating.

---

> > > ### Author Response · Authors · 2024-11-27
> > >
> > > Dear reviewer, thanks again for your feedback on our manuscript. Based on all of the feedback we received, we decided to revise our introduction in another version of our manuscript. We have uploaded it for you to read. The latest version of our manuscript still includes changes that address your concerns but does not highlight them anymore. We decided to do so such that reading our manuscript is easier for your final rating.
> > >
> > > We would also like to kindly remind you to increase your rating as you mentioned in your last comment.

---

### Official Review · Reviewer_jr7d · 2024-11-01

**Soundness:** 3
**Presentation:** 3
**Contribution:** 3
**Rating:** 6
**Confidence:** 4

**Summary:**

The paper introduces a new dataset in the field of computational ethology. It focuses on creating various scenarios where mouse behaviour and actions are recorded while it tries to solve complex mechanical lockbox opening problems in return for treats. The novelty of the paper lies with the fact that existing datasets in this topic focuses only on problems which are not sufficiently complex or are concerned with studying social behaviour of animals.

**Strengths:**

•	The dataset has over 117 hours of videos of mice behaviour doing complex tasks. This multi-perspective video data is novel and crucial to enable the development of advance methods in the field of neuroscience.
•	Action based labelling is very detailed, providing annotations at sub-second intervals.
•	The authors are very transparent in how they gathered the data and in compliance of local laws regarding to animal research.
•	Human annotation is provided by the authors with special care taken to ensure that there is no leakage between the labelled and unlabelled set by keeping the mouses separate for labelled set.
•	The human annotators followed a specific set of ethograms which makes the process objective in nature and lowers the possibility of disagreement.
•	The authors provide detailed statistics of their dataset

**Weaknesses:**

•	Labelled data provided is very limited and represents only 13% of the total available data. More annotated data would be immensely helpful to researchers.
•	The dataset focuses exclusively on female mice and the reasoning behind this is not explained and the authors do not cite any paper which disproves this. This choice could limit generalizability.
•	There is an uneven distribution between playtime action labels. If this cannot be made closer to each other, the authors should mention why it is so.
•	The dataset is provided in grayscale. While this is not a major weakness, RGB images could contain more information that is lost in the grayscale conversion.
•	The authors do not provide camera homography information of the cameras involved.

**Questions:**

•	It would be great if the authors provided a baseline method on a task with their dataset.
•	A data dictionary of the labels as well as a getting started notebook was not provided. These would greatly increase the ease of use of the dataset.

---

> ### Author Response · Authors · 2024-11-21
>
> Dear reviewer, we are thankful for your insightful feedback as it allowed us to improve our manuscript. Please allow us to address both the weaknesses and questions you stated in your review.
>
> Any changes that we have introduced to the text of our revised manuscript have been marked with blue color.
>
> As weakness 1, you argue that providing behavior labels for a larger split of our dataset  would be helpful. We agree. However, our goal is not to provide an exhaustively labeled dataset for supervised learning, but to contribute to the ongoing shift towards unsupervised methods by providing a high-quality test set. We provide more detailed action labels than most other datasets (i.e., 5 interactions x 4 mechanisms) and assess their quality (i.e., inter-rater reliability) systematically. This resonates with the ongoing debate in ML on whether to prefer larger amounts of noisy or smaller amounts of precise label data. Providing substantially more high-quality label data is currently beyond our capacity.
>
> As weakness 2, you point out that our data exclusively shows female mice and that this may bias the analysis. We agree. However, the mice are group-housed and male groups often suffer from aggressive conflicts, while mixed groups come with their own challenges (e.g., growing number of animals). These challenges may bias the analysis as well.
>
> As weakness 3, you point out that the labels are not distributed evenly. This is rooted in the mice behaving freely in the arena. Their preference for different actions and mechanisms is naturally occurring and reflected in the statistics we report. We have added this brief explanation to our manuscript. We have also added an “any” column to Tbl. 4 describing how often each of the action labels is active in total, and fixed a mistake in our calculations.
>
> As weakness 4, you criticize that we provide grayscale videos. That’s the result of us recording in infrared. Infrared light is invisible to the mice. Recording in RGB requires bright light sources in the visible spectrum of the mice, which is often aversive to mice and biases their behavior. This is a standard approach when conducting animal experiments in controlled environments. We added a brief explanation to the first paragraph of section 3.1.
>
> As weakness 5, you point out that we do not provide homography parameters. We unfortunately did not record them. But we aim to mitigate this problem as we will provide the lockboxes’ CAD model files, which provide ground truth anchor points for their keypoints. We will provide these files upon acceptance as they could possibly deanonymize us.
>
> In question 1, you asked us to provide a baseline method. While there exist methods for behavior identification on individual mice (e.g., Keypoint-MoSeq [1], B-SOiD [2], and VAME [3]), these methods extract behavioral primitives of a mouse in isolation, rather than interactions with its environment as we do here. It is nontrivial to extend them such that they couple an action (e.g., touch) with an environmental element (e.g., a lockbox mechanism) and there is currently no established standard that could serve as a benchmark. Therefore, we decided against using these existing methods as a benchmark for our dataset. Instead, we applied an established keypoint tracking method (i.e., DeepLabCut [4]) to extract the 2D keypoints from the videos and implemented a generic processing pipeline, based on (Extended) Kalman filters, to detect proximity, touching, and biting in 3D space. We describe this method and evaluate its results in sections 2.1 and 3.4.
>
> In question 2, you asked us to add a label dictionary and an exemplary notebook. We have added the label-dictionary.txt file to our dataset and its preview. We have considered providing a notebook with our submission, but decided against it. Reason being that our data can be applied with a broad spectrum of contexts (e.g., keypoints vs abstract representations, machine learning vs neuroscience) and we could not think of a way to provide a universal notebook.
>
> We hope that we were able to properly address your questions and remarks, and appreciate additional feedback.
>
> [1] Weinreb et al. Keypoint-MoSeq: parsing behavior by linking point tracking to pose dynamics. Nature Methods, vol. 21, no. 7, pp. 1329–1339, 2024.
>
> [2] Hsu et al. B-SOiD: an open source unsupervised algorithm for discovery of spontaneous behaviors. BioRxiv, 770271, 2019.
>
> [3] Luxem et al. Identifying behavioral structure from deep variational embeddings of animal motion. Communications Biology, vol. 5, no. 1, pp. 1267, 2022.
>
> [4] Mathis et al. DeepLabCut: markerless pose estimation of user-defined body parts with deep learning. Nature Neuroscience, vol. 21, pp. 1281–1289, 2018.

---

> > ### Author Response · Authors · 2024-11-27
> >
> > Dear reviewer, based on the discussion with the other reviewers we decided to revise our introduction again. The latest version of our manuscript still includes changes that address your concerns but does not highlight them anymore. We decided to do so such that reading our manuscript is easier for your final rating.

---

### Official Review · Reviewer_562X · 2024-11-02

**Soundness:** 3
**Presentation:** 2
**Contribution:** 3
**Rating:** 6
**Confidence:** 4

**Summary:**

The paper introduces a novel video dataset of individual mice interacting with complex mechanical lockbox puzzles, labeled as the "Mouse Lockbox Dataset." It provides a significant contribution to computational neuroethology, presenting over 110 hours of video data capturing intricate, non-social behaviors. The dataset uniquely includes multi-perspective recordings, human-annotated action labels for a subset of videos, and benchmark results utilizing a keypoint-based action classification method. This work aims to support machine learning advancements in behavioral action classification, particularly in fine-grained tasks such as object manipulation.

**Strengths:**

1. This dataset addresses a crucial gap in animal behavior research by focusing on complex, single-agent interactions in mice—particularly rare for capturing intricate, non-social behaviors.
2. Its scale and detail are impressive, with over 110 hours of footage from multiple angles. The multi-perspective setup and extensive annotations make it a highly valuable resource for training and evaluating behavior recognition models.
3. The inclusion of a keypoint-based action classification benchmark is a well-considered addition. It establishes a standard for future comparisons and highlights the strengths of current methods (e.g., proximity detection) as well as their limitations (e.g., recognizing fine-grained actions).
4. The authors provide a detailed methodology for creating the dataset, which supports reproducibility and helps ensure the reliability of the process.

**Weaknesses:**

1. Although this dataset is a valuable resource for behavior recognition, the paper lacks clear examples of specific use cases and potential research problems it could address. The absence of concrete applications or suggested research directions may limit its utility as an inspiration for future work. Providing examples of problems or experimental scenarios where this dataset could be useful would strengthen its research impact and relevance.
2. The pseudo-synchronization of the multi-camera setup presents a minor challenge for users needing precise 3D reconstructions. While the misalignment is slight, it may still affect analyses requiring high precision.
3. Certain actions, such as biting, have lower annotator agreement, indicating challenges in reliably annotating these behaviors. This suggests that alternative or more standardized annotation methods may be necessary to improve reliability.

**Questions:**

1. Could you clarify whether alternative pose-tracking methods were considered, particularly to improve the accuracy of detecting fine-grained actions like biting?
2. Is there a plan to expand the dataset with additional annotated videos, or are any tools being developed to help automate or verify annotation accuracy for challenging actions?
3. Given the temporal desynchronization in the multi-perspective recordings, have you considered any post-processing or alignment correction methods to reduce this effect?

---

> ### Author Response · Authors · 2024-11-21
>
> Dear reviewer, we are thankful for your insightful feedback as it allowed us to improve our manuscript. Please allow us to address both the weaknesses and questions you stated in your review.
>
> Any changes that we have introduced to the text of our revised manuscript have been marked with blue color.
>
> As weakness 1, you criticize the lack of concrete use cases and research questions that our dataset is relevant for. We have added a short paragraph to the introduction, in which we outline how action labels enable computational analyses of complex behaviors. While we acknowledge the charm of giving a granular analysis of both its potentials and the literature available today, we refrained from doing so as this could otherwise distract readers from the core contribution of our work.
>
> A weakness 2 and question 3, you point out that the temporal pseudo-synchronization that we also report on in section 3.5 may be an additional challenge in some contexts. We have considered post-hoc synchronizing the videos. However, we consider the temporal desynchronization of 1.39±1.5 frames in our data not critical given the temporal accuracy of ±3 frames of our behavioral labels. We could also not think of any method that would produce reliable results without substantial amounts of labor. Given this cost-benefit trade-off, we decided not to alter the raw data.
>
> As weakness 3, you point out that the inter-rater reliability of the biting label is lower than for other labels and suggest that we use more standardized annotation methods. We would like to emphasize that we follow best-practise for labeling. Reporting the inter-rater reliability is the gold standard, but unfortunately the vast majority of work that presents new data falls short to do so. We also follow the best practice by having exclusively skilled human raters annotate the data that also have been specifically instructed prior to working with our data. The low inter-rater reliability is the mere reflection that biting is hard to detect even for expert human raters, e.g., due to occlusions. We hope that both the reviewers and readers of our work appreciate our transparency regarding the low inter-rater reliability of a subset of our labels.
>
> In question 1, you ask us whether alternative post-processing methods were considered to improve our baseline method’s accuracy. We are sure that the results can be improved by specialized and fine-tuned methods. Our benchmark rather reflects the current performance of current standard methods. Proposing a new state-of-the-art algorithm for action classification is not the goal of the paper. We are looking forward to seeing what the field will develop.
>
> In question 2, you ask us if we are planning to extend the dataset and if there are approaches under development that would further automate behavior annotation. We agree that providing more behavior labels would be beneficial in some contexts. However, our main goal was not to provide an exhaustively labeled dataset for supervised learning, but rather contribute to the ongoing shift towards self- and unsupervised methods by providing a high-quality test set. To this end, we provide more detailed action labels than most other datasets (i.e., 5 interactions x 4 lockbox components + reward consumption) and estimate their quality (i.e., inter-rater reliability) systematically. This resonates with the ongoing debate in ML on whether to prefer larger amounts of noisy or smaller amounts of precise label data. Labeling a substantially larger fraction of the videos at this quality is currently beyond our capacity. As we stated above, proposing a new state-of-the-art algorithm for action classification is not the goal of the paper.
>
> We hope that we were able to properly address your questions and remarks, and appreciate additional feedback.

---

> ### Comment · Reviewer_562X · 2024-11-22
>
> Thank you for your detailed response, which has addressed my concerns. I am willing to raise the score for this paper.

---

> > ### Author Response · Authors · 2024-11-27
> >
> > Dear reviewer, thanks again for your feedback on our manuscript. Based on all of the feedback we received, we decided to revise our introduction in another version of our manuscript. We have uploaded it for you to read. The latest version of our manuscript still includes changes that address your concerns but does not highlight them anymore. We decided to do so such that reading our manuscript is easier for your final rating.

---

### Official Review · Reviewer_TrHr · 2024-11-04

**Soundness:** 3
**Presentation:** 3
**Contribution:** 2
**Rating:** 5
**Confidence:** 3

**Summary:**

The authors propose a novel dataset, the _Mouse Lockbox Dataset_, which includes videos of individual mice dealing with mechanical puzzles named lockboxes. While existing datasets mainly target mice's social or common behaviors, the aim of the proposed dataset is to provide a useful resource for studying single-agent intelligent behaviors.
The paper describes with great detail both the setup used by the authors to record it and the procedure they established to annotate the (13% of the) dataset. The human-annotated labels are also compared with a benchmark automatic method, which relies on the poses of the mice as extracted with DeepLabCut. In particular, the authors compare the human-human agreement on the labels with the one obtained by comparing the human annotator and the automatic method performance.

**Strengths:**

1. The paper is clear and well-written, reporting several important details on the dataset generation process.
2. The dataset represents an interesting source for computational ethology to study the mice's behavior in controlled environments when dealing with mechanical problem-solving.
3. The ~13% of the dataset is annotated with several features that can be leveraged to effectively delve into the mice's behavior.
4. The related work section clearly reports and compares the proposed dataset with the most important datasets in rodent computational ethology literature.

**Weaknesses:**

1. While the proposed dataset represents a novel source for studying mice behavior, the authors lack a proper experiment to showcase the relevance of their contribution. Benchmarking an automatic method against the human annotators reinforces the idea of a high-quality dataset but does not deliver to a non-ethologist reader the idea of how to use the  _Mouse Lockbox Dataset_.
2. To complete section 2, the authors could report a table comparing their proposed dataset with the available literature described therein.
3. The method used for benchmarking is not reported in the paper, although it is cited in the introduction. To help the reader understand the importance of this baseline, the authors should introduce it in its dedicated section, reporting at least some high-level details about the structure of this baseline.
4. The introduction is fragmented (with several paragraphs) and does not focus enough on the paper's contributions. In particular, the section describes recent advancements and methods in computational ethology well, but only the last two paragraphs describe the content of the proposed paper.
5. In general, the paper would greatly benefit from introducing and evaluating some methods that can be used on the _Mouse Lockbox Dataset_ to provide a reference for future users and encourage research with it.

**Questions:**

1. Will the key-point tracking results be released with the _Mouse Lockbox Dataset_? In the conclusive section, the authors claim, "(...) we are convinced that approaches beyond keypoint (pose) tracking, e.g., representations learnt without any or under self-supervision, are crucial to future advancements in neuroscience.", however when dealing with metric tasks, e.g., estimating the distance of the mouse from the lockbox, keypoint data represent an important piece of information. Also, to qualitatively assess the baseline performance, adding an image of the key points extracted by the baseline would be interesting.

---

> ### Author Response · Authors · 2024-11-21
>
> Dear reviewer, we are thankful for your insightful feedback as it allowed us to improve our manuscript. Please allow us to address both the weaknesses and questions you stated in your review.
>
> Any changes that we have introduced to the text of our revised manuscript have been marked with blue color.
>
> As weaknesses 1 and 4, you argue that we lack concrete use cases and research questions that our dataset is relevant for and the fragmentation of our introduction. We have added a short paragraph to the introduction, in which we outline how action labels enable computational analyses of complex behaviors. And while we acknowledge the charm of giving a granular analysis of both its potentials and the literature available today, we refrained from doing so as this could otherwise distract readers from the core contribution of our work.
>
> As weakness 2, you criticize the lack of a table comparing our dataset against others. We have added such a table (Tbl. 1).
>
> As weakness 3, you point out that the benchmark method is not well described in the manuscript. We have now extended its description in section 2.1.
>
> As weakness 5, you point out that our work would benefit from introducing additional baseline methods. While there exist methods for behavior identification on individual mice (e.g., Keypoint-MoSeq [1], B-SOiD [2], and VAME [3]), these methods extract behavioral primitives of a mouse in isolation, rather than interactions with its environment as we do here. It is nontrivial to extend them such that they couple an action (e.g., touch) with an environmental element (e.g., a lockbox mechanism) and there is currently no established standard that could serve as a benchmark. Therefore, we decided against using these existing methods as a benchmark for our dataset. Instead, we applied an established keypoint tracking method (i.e., DeepLabCut [4]) to extract the 2D keypoints from the videos and implemented a generic processing pipeline, based on (Extended) Kalman filters, to detect proximity, touching, and biting in 3D space. We describe this method and evaluate its results in sections 2.1 and 3.4.
>
> In your question, you ask us if we will release the keypoint tracking results alongside our dataset. Yes, we will also publish the raw DeepLabCut tracks for all three perspectives for the entire dataset, so that users can freely decide on their methods for 3D triangulation, smoothing etc. There, we hope to make the dataset attractive also to researchers that wish to get started with a keypoint-based approach.
>
> We hope that we were able to properly address your questions and remarks, and appreciate additional feedback.
>
> [1] Weinreb et al. Keypoint-MoSeq: parsing behavior by linking point tracking to pose dynamics. Nature Methods, vol. 21, no. 7, pp. 1329–1339, 2024.
>
> [2] Hsu et al. B-SOiD: an open source unsupervised algorithm for discovery of spontaneous behaviors. BioRxiv, 770271, 2019.
>
> [3] Luxem et al. Identifying behavioral structure from deep variational embeddings of animal motion. Communications Biology, vol. 5, no. 1, pp. 1267, 2022.
>
> [4] Mathis et al. DeepLabCut: markerless pose estimation of user-defined body parts with deep learning. Nature Neuroscience, vol. 21, pp. 1281–1289, 2018.

---

> > ### Comment · Reviewer_TrHr · 2024-11-21
> >
> > Dear authors,
> >
> > I appreciate the major revisions the paper underwent, and I think some points are definitely more clear now.
> >
> > In particular, Table 1 greatly helps the reader to understand the differences between sources focusing on rodent behaviors at a glance, although I have some more feedback to improve that table:
> > 1. The layout of the table does not look very nice. I would suggest the authors use the borders for separating columns and rows
> > 2. Instead of writing "# PERSPECTIVE x REAL TIME" or "CONTEXT", use different columns (e.g., "Dataset", "# Perspectives", "Recording Total time", "Rodent family", "Context") and report the best numbers in boldface
> >
> > As for these suggestions, the authors could refer to, e.g., Table 1 in [Patel et al., 2021].
> >
> > As for weakness 3, I read and appreciate the extended explanation.
> >
> > On the other hand, I did not appreciate the answers to the other weaknesses/questions.
> >
> > As for weakness 1, I did not understand why the authors coupled it together with weakness 4. In my opinion, weakness 1 is linked with weakness 5 instead, as they both require the authors to perform some experiments on their proposed dataset to provide the community with reference performance and kickstart the research with it. This is actually very important; indeed, it has been remarked on by **all** the reviewers in this session. Copypasting a single answer to all of the reviewers is not enough, and still think the lack of presentation of experiments showcasing tasks/baselines/performance is a critical weakness, which leads me towards a rejection. I do not think that readers would be "distracted" from the contributions of the work in this way; instead, they would get more engaged and could start from a standardized codebase or at least some performance indicators.
> >
> > Moreover, (weakness 4) I read the updated version of the introduction. It is still not focusing on why the community needs the proposed dataset ("absolute" (importance of computational ethology with mice) and "relative" (differences from similar datasets) motivations), even though the added paragraph is a valuable addition. The first 6/9 paragraphs in the introduction mostly focus on providing the readers with elements and insights into computational ethology, which, in my view, further separate the reader from the paper's contribution. To have an example of what I am suggesting, the authors could again refer to [Patel et al., 2021], and they should note that in their introduction, the authors do not spend many words explaining Human Mesh Recovery but directly bring motivations for their source and showcase it and the content of the paper.
> >
> > [Patel et al. 2021]: Patel *et al.*, "AGORA: Avatars in Geography Optimized for Regression Analysis", CVPR 2021.
> >
> > Note: I refer to [Patel et al. 2021] as an example, but, as reviewer mH4b pointed out, there is a kind of standard for the "dataset papers", so several other works can be referenced as well.

---

> > > ### Author Response · Authors · 2024-11-27
> > >
> > > Dear reviewer, thanks for the clarification. It allowed us to better understand your criticism. We did a major revision to our introduction and uploaded the latest version of our manuscript for you to read. The latest version of our manuscript still includes changes that address your concerns but does not highlight them anymore. We decided to do so such that reading our manuscript is easier for your final rating.

---

> > > > ### Comment · Reviewer_TrHr · 2024-11-29
> > > >
> > > > Dear authors,
> > > >
> > > > I appreciate the significant revision to the Introduction. In its current state, the Introduction explains well the connections between (computational) Ethology and broader fields, such as Robotics or Representation Learning, motivating and encouraging the use of this kind of dataset. Also, the second and third paragraphs directly point the reader to the unique features of the proposed dataset (even though I would further suggest providing some examples after the sentence *"However, all (...) trivial and social behaviors"*) in a positive manner, proposing research directions which are possible to follow after the *Mouse Lockbox Dataset*, which is valuable.
> > > >
> > > >
> > > > On the other hand, I agree with reviewer mH4b that the authors must do more work to demonstrate the actual significance of the proposed resource, which can not only be expressed through words. Instead, it should be confirmed by experiments that quantitatively describe your proposal and introduce the tasks a researcher can investigate using the *Mouse Lockbox Dataset*.
> > > > Indeed, as the authors describe the importance of exploring new directions for computational ethology (and I probably agree), claiming the proposed *Mouse Lockbox Dataset* is a relevant source, they should also provide some experiments and baselines, motivating the reader to follow the suggested path (see, e.g., first and third bullet points in mH4b follow-up).
> > > >
> > > > As a final remark, while I acknowledge the authors consider some of my and other reviewers' suggestions (as confirmed by their last iteration of the manuscript), I also note that in their last comment, they overlooked some of my concerns entirely: the suggestions about Table 1, and the remarks on weaknesses 1 and 5 in my first review. I understand that the authors can have different opinions and do not want to, for instance, modify Table 1 for some reason. Still, I would expect at least a comment on their follow-up rebuttal motivating their choices and arguing against my suggestions.
> > > >
> > > > Since I consider the lack of experiments a critical weakness and the authors' answers did not address my concerns, I will reconsider my score.

---

> > > > > ### Author Response · Authors · 2024-11-29
> > > > >
> > > > > Dear reviewer, thanks for ongoing engagement in the rebuttal and your valuable feedback. We want to apologize for our insufficient communication. We were under the impression to have responded to all of the weaknesses of your initial review.
> > > > >
> > > > > Regarding the additional changes you suggested for the setup of table 1, we have decided against them as we don’t think they’d increase its descriptiveness. Instead, we want to keep the table simple and avoid confusion regarding the playtimes.
> > > > >
> > > > > Regarding your initial weakness 1, we tried our best to better explain the use cases of our dataset in the revised introduction of our manuscript. Please let us know if we still have not managed to properly express ourselves.
> > > > >
> > > > > Regarding your initial weakness 5, the approaches available today are designed for trivial and social behaviors but neglect intricate agent-object interactions. Their application on our dataset would require modifications that prevent direct comparison of their performance against that of their originally intended use cases.
> > > > >
> > > > > The remarks above are also stated in our latest reply to reviewer mH4b.

---

### Official Review · Reviewer_mH4b · 2024-11-04

**Soundness:** 3
**Presentation:** 2
**Contribution:** 3
**Rating:** 5
**Confidence:** 4

**Summary:**

The authors present a new, large dataset of lab mice solving mechanical lockbox puzzles. Unlike existing animal behavior datasets, their lockboxes capture more cognitive / learning / problem solving abilities of the animals. In the future, this dataset, or the use of the lockboxes morie generally, could be used to better understand motor and cognitive learning and complex neural computations and processes. At the moment, the dataset's benchmark includes recognizing a set of actions or action.puzzle states.

**Strengths:**

The author present an interesting and novel new dataset of mice solving puzzles. It is not 100% clear to me the extent of the research questions one could use such a dataset for, and I encourage the reviewers to expound on this in their revision.

**Weaknesses:**

As a dataset paper, the authors should be providing additional technical details, data quality metrics, and an expanded set of methods for benchmarking. I've left specific comments on these points in the questions section.

**Questions:**

A few things in the introduction that would be good to clarify in the text.

L55 -- you say that it is a disadvantage that poses are low dimensional. But there are many scenarios where this is advantage.
L58 -- I don't understand what you mean when you write, "It impedes mapping of keypoint locations from different perspectives into a common coordinate system."
L59-60 -- there is actually an ideal set of keypoint locations -- skeletal joints
L107 -- "the more recent works" are actually older than some of the ones you cite in the previous sentence

Relatedly, you spend a lot of the intro saying how bad it is to use keypoints for behavioral analysis, but then go on to use keypoints for your benchmark action recognition results. What am I missing?

Please provide a clear breakdown of how much of your dataset video time is the animals engaged with the puzzle. What is your definition of "playtime"? I would prefer if these data were in the table and not in a ring/pie chart that is harder to read.

How precise is your keypoint tracking (3D error in mm)? How many keypoints were tracked exactly? What tracking method did you use? What method/model did you use to go from keypoints to frame labels in your benchmarking? There is a lot of missing detail that is required of a datasets / benchmarks paper that the authors must provide for the work to be useful to others.

Similarly, as a dataset/benchmark paper, it is standard to show results from several different methods, or at least some internal comparisons over different model configurations/hyperparamters.

---

> ### Author Response · Authors · 2024-11-21
>
> Dear reviewer, we are thankful for your insightful feedback as it allowed us to improve our manuscript. Please allow us to address both the weaknesses and questions you stated in your review.
>
> Any changes that we have introduced to the text of our revised manuscript have been marked with blue color.
>
> You mention that the use cases and potentially addressed research questions of our datasets are not entirely clear to you. The new paragraph in lines 052–057 outlines how action labels enable computational analyses of complex behaviors. We acknowledge the charm of reviewing applications in the diverse literature available today, but refrained from doing so as this could distract readers from the core contribution of our work.
>
> ➝ L55 -- you say that it is a disadvantage that poses are low dimensional. But there are many scenarios where this is advantage.
>
> ➝ L59-60 -- there is actually an ideal set of keypoint locations -- skeletal joints
>
> ➝ Relatedly, you spend a lot of the intro saying how bad it is to use keypoints for behavioral analysis, but then go on to use keypoints for your benchmark action recognition results. What am I missing?
>
> By submitting to ICLR we hope to fuel the emerging adoption of representation learning in computational ethology. That is why we point out challenges of keypoint-based methods, although they are undoubtedly state-of-the-art. Methods for behavioral studies that rely on representation learning are—despite their potentials—currently under-researched. Therefore, keypoint methods remain the only reasonable benchmark. To adequately portrait keypoint-based methods we changed our phrasing in lines 057 and 060.
>
> ➝ L58 -- I don't understand what you mean when you write, "It impedes mapping of keypoint locations from different perspectives into a common coordinate system."
>
> We have rephrased our statement to be more precise and clear.
>
> ➝ Please provide a clear breakdown of how much of your dataset video time is the animals engaged with the puzzle.
>
> We have added an “any” column to Tbl. 4 describing how often each of the action labels is active in total. We have furthermore fixed a mistake in our calculations and updated the values in this table.
>
> ➝ What is your definition of "playtime"? I would prefer if these data were in the table and not in a ring/pie chart that is harder to read.
>
> We have added our definition of total playtime, the number of perspectives multiplied by the real time recorded, to our manuscript. We have also added a new table (Tbl. 1) to compare other datasets against ours.
>
> ➝ How precise is your keypoint tracking (3D error in mm)?
>
> ➝ How many keypoints were tracked exactly?
>
> We added Fig. 5 to our appendix A.2 showcasing the keypoints we used. There we also report our tracker’s 2D training and test errors.
>
> ➝ What tracking method did you use?
>
> ➝ What method/model did you use to go from keypoints to frame labels in your benchmarking?
>
> → There is a lot of missing detail that is required of a datasets / benchmarks paper that the authors must provide for the work to be useful to others.
>
> ➝ Similarly, as a dataset/benchmark paper, it is standard to show results from several different methods, or at least some internal comparisons over different model configurations/hyperparamters.
>
> We have now extended the description of the used benchmark used in section 2.1.
>
> While there exist methods for behavior identification on individual mice (e.g., Keypoint-MoSeq [1], B-SOiD [2], and VAME [3]), these methods extract behavioral primitives of a mouse in isolation, rather than interactions with its environment as we do here. It is nontrivial to extend them such that they couple an action (e.g., touch) with an environmental element (e.g., a lockbox mechanism) and there is currently no established standard that could serve as a benchmark. Therefore, we decided against using these existing methods as a benchmark for our dataset. Instead, we applied an established keypoint tracking method (i.e., DeepLabCut [4]) to extract the 2D keypoints from the videos and implemented a generic processing pipeline, based on (Extended) Kalman filters, to detect proximity, touching, and biting in 3D space. We describe this method and evaluate its results in sections 2.1 and 3.4.
>
> We hope that we were able to properly address your questions and remarks, and appreciate additional feedback.
>
> [1] Weinreb et al. Keypoint-MoSeq: parsing behavior by linking point tracking to pose dynamics. Nature Methods, vol. 21, no. 7, pp. 1329–1339, 2024.
>
> [2] Hsu et al. B-SOiD: an open source unsupervised algorithm for discovery of spontaneous behaviors. BioRxiv, 770271, 2019.
>
> [3] Luxem et al. Identifying behavioral structure from deep variational embeddings of animal motion. Communications Biology, vol. 5, no. 1, pp. 1267, 2022.
>
> [4] Mathis et al. DeepLabCut: markerless pose estimation of user-defined body parts with deep learning. Nature Neuroscience, vol. 21, pp. 1281–1289, 2018.

---

> > ### Author Response · Authors · 2024-11-27
> >
> > Dear reviewer, based on the discussion with the other reviewers we decided to revise our introduction again. The latest version of our manuscript still includes changes that address your concerns but does not highlight them anymore. We decided to do so such that reading our manuscript is easier for your final rating.

---

> > ### Comment · Reviewer_mH4b · 2024-11-28
> >
> > I appreciate the changes the authors have made to the introduction, and the inclusion of additional technical detail is a step in the right direction.
> >
> > As I said in my initial review, I do think this dataset is unique and has the potential to be valuable. However, I think the authors still have some work to do to craft a paper that clearly establishes the value of the dataset and lays a foundation for the community, through experiments and analyses, not just more writing.
> >
> > For more explanation of what I mean, look to the papers presenting the datasets the authors have compared to in Table 1. All of these papers go beyond merely describing a dataset, its annotations, and a single simple baseline. They design and test new architectures and algorithms and compare between them, they explore the effects of hyperparameter choices, they crunch and plot and dig into the signals and patterns in the data. In some cases, they present fundamentally new modes of analysis.
> >
> > Examples of extensions that align with the paper’s stated motivations and would elevate the study:
> >
> > - The authors state that we should actually be focusing on unsupervised representation learning, or if supervised then at least we should ultimately move away from keypoint-based analyses. Exploring these ideas would not be too onerous, in fact there are already a variety of existing approaches they could implement or extend for this. Classifying animal behavior directly from video is not uncommon (e.g., DeepEthogram -- Bohnslav et al. 2021). There is also work on unsupervised representation learning for mouse behavior (e.g. BehaveNet – Batty et al. 2019). And there are many studies using 2D or 3D keypoints for unsupervised analysis of behavior (some of these papers the authors cite, e.g. Dunn et al. 2021). For example, using keypoint-based approaches, the authors could identify  behavioral motifs and examine whether there are any interesting relationships between mechanism type, lock state, or proximity.
> >
> > -	The authors speculate that the poor performance of touch and biting classification is due to imprecision when detecting distances from keypoints to bounding boxes, or potentially that thresholds or sizes used for detecting proximity need to be better tuned. This could (and should) be explored. Likely using a very simple classifier based on a single keypoint, as the authors do, is not sufficient – it could be explored whether it helps to take into account other pose features captured by the other keypoints on the body. Classifiers that take in postural time series could be tested, etc.
> > -	An analysis or some kind that conveys concretely what the authors have in mind when they say their keypoint benchmark will “contribute to the ongoing shift towards self- and unsupervised methods by providing a high-quality test set”.
> >
> > Other comments/concerns:
> >
> > 1.	The authors have had the opportunity to address reviewer TrHr’s well-articulated follow-up response regarding the need for additional experiments (a concern I also share), but have not done so. Also, simple suggestions made by TrHr to clean up Table 1 appear to have been ignored.
> > 2.	The authors stated that Appendix A.2 contained 2D tracking performance metrics, but these measures are not there.
> > 3.	As the authors are using 3D keypoints for their analyses, their error metrics should really be in 3D, not in 2D.
> > 4.	The authors highlight multiple times that they provide 33% more video than any existing public mouse dataset. But this is only because they multiply their hours by the number of camera perspectives, which seems contrived. Taken to an extreme, what if someone used 1000 cameras and recorded only 10 minutes of behavior total. Is it meaningful to say this would be the largest dataset of mouse behavior? To me, the more important metric is the amount of animal behavior recorded (independent of perspective count). If anything, the social datasets should get another multiplier x2 to make it total mouse hours in the datasets (which would then make CalMS21 and CRIM13 larger than lockbox).
> > 5.	‘Intelligent’ behavior is a loaded and controversial term. Social interactions certainly require intelligence, and they also involve interactions with other entities in the environment.
> > 6.	Social datasets also have interaction behavior labels, and they introduce techniques for identifying behaviors that involve interactions with their environment. So I disagree with the statement that existing methods for behavioral identification “extract behavioral primitives of a mouse in isolation, rather than interactions with its environment as we do here.”

---

> > > ### Author Response · Authors · 2024-11-29
> > >
> > > Dear reviewer, we appreciate your acknowledgment of our work’s potential and your feedback. However, we do not appreciate that you refrained from actively participating in the rebuttal and instead provided such elaborate additional feedback, more detailed than your initial review, only after the rebuttal period ended.
> > >
> > > We still want to respond to your latest comment.
> > >
> > > We deliberately decided not to introduce a new methodology alongside our dataset. As we have explained before, we intend to enable the community to work on interesting, previously inaccessible problems, not to attempt to solve them. We further believe that separating methodology from data helps the community better distinguish different contributions. We agree this isn’t standard practice, but we still think this is as it should be.
> > >
> > > We acknowledge unsupervised approaches in the field and separate them from others in our manuscript. However, these approaches are designed for trivial and social behaviors, not intricate agent-object interactions. Therefore, applying them to our dataset requires modifications that may make performance comparisons questionable. Further, reliable action classification—the foundation for behavioral analysis—remains unsolved. In that sense, suggesting that we could have easily conducted behavioral analysis by mere application of existing approaches is incorrect.
> > >
> > > Regarding your last remark before your additional concerns, we don’t consider our keypoint-based benchmark to contribute to the shift towards self- and unsupervised methods. After rereading our manuscript and the rebuttal comments, we can’t find where this impression comes from. Maybe there’s an interpretation error in the sections where we hope our dataset will serve this purpose.
> > >
> > > Regarding your other concerns:
> > >
> > > 1. We addressed the feedback to provide a better overview of the related work. The well-placed feedback led to adding a table with relevant information. Reviewer TrHr further suggested a table setup change, but we decided against it as it wouldn’t enhance informativeness. We kept the table simple to avoid clutter and confusion regarding playtimes.
> > >
> > > 2. We provided 2D tracking information metrics in Appendix A.2 on page 14. You likely overlooked it or read a previous manuscript revision.
> > >
> > > 3. Since we track 2D keypoints using DLC, as is standard practice, we only have perspective-wise 2D ground-truth keypoint labels for the mice. While one could report 3D error for the lockbox mechanisms based on their CAD models, this would mislead readers, as the challenge lies in tracking the highly dynamical mouse rather than the barely moving lockbox.
> > >
> > > 4. We’re transparent about our playtime reporting, distinguishing between total playtime and recorded real time. We share your concern about confusion, but as is evident from reading the other rodent datasets from our related work section, (a) reporting total playtime is common practice and (b) our reporting is more rigorous and transparent than anyone else’s. We reject the accusation of counterfeiting these metrics. We further believe it’s more nuanced than your example of 1000 cameras. Multiplying 1000 cameras with 10 minutes of playtime is absurd, but providing video footage from a reasonable amount of perspectives offers a clear benefit over a single view.
> > >
> > > 5. We acknowledge the ambiguity of the term “intelligent” and explain its meaning in our manuscript. We also clarify the distinction between intelligent behavior in our dataset and other rodent datasets by listing all annotated behavior labels in the related work section. In our dataset, intelligent behavior refers to mice learning to solve mechanical problems.
> > >
> > > 6. We would appreciate it if you provide concrete examples of approaches directly applicable to our dataset, that solve tasks where precise localization (e.g., touch) rather than relative localization (e.g., chase) is required, as these are qualitatively different problems. We’re not aware of any.

---

### Meta-Review · Area_Chair_9PS1 · 2024-12-20

**Metareview:**

This paper introduces a new dataset consisting of mice solving puzzles (lockboxes), captured over 40 hours from 3 camera views. There are 12 total mice in the dataset, and 2/12 has human-annotated behavior labels (6 total labels). Each video has 1 mouse, and videos have been manually trimmed to remove experimenter hands coming in/out. The authors test a baseline method on their dataset, which is to use keypoint proximity to compute behavior labels. While reviewers appreciate the introduction of a new dataset, there are concerns remaining on the experimental design, lack of benchmark models on the dataset, and some missing details on the baseline. It appears that many standard baselines and analysis are missing (non-exhaustive list of example experiments to conside: clustering using keypoints, different keypoint definitions, different keypoint estimators, different hand-crafted features, image/video embeddings -- standard ones like CLIP or DINO would work). The AC encourage the authors to refer to other similar datasets published at ML conferences for the expectations on using experiments to illustrate the contribution of the dataset (see CalMS21, PAIR-24M, MABe22 (ICML 2023), ChimpACT (NeurIPS dataset and benchmarks 2023)). Separately, the AC would like to note that it is unusual in the field to multiply the hour of recorded videos by the # of views -- the goal is not to have the largest amount of hours, but to provide a dataset that can provide the community with interesting insights. Reviewers have left valuable feedback for this paper, and taking those into account could help the authors make this dataset a strong contribution to the community.

**Additional Comments On Reviewer Discussion:**

The key concern raised by reviewers include clarifications on the baseline approach (keypoint proximity) and motivation, as well as, crucially, lack of methods evaluated on the dataset (to help demonstrate the contribution of the dataset to the community). Some concerns were addressed during the rebuttal session, but core concerns on the experiments remain unaddressed. In particular, there are many standard approaches for keypoints and image/video representation learning (some of which have already been applied to behavior analysis (see papers linked above), although not specifically on the new Mouse Lockbox dataset, it is important to show how this dataset can provide additional insights by running similar experiments to demonstrate gaps). There are also well-used models for computing visual representations (e.g. CLIP and DINO) which are not discussed or used in this work. These missing experiments and analysis demonstrating how the dataset fits into the current state of the community is a key concern.

---

### Decision · Program_Chairs · 2025-01-22

Reject